# An Evaluation of Jaw Tracking Movements in Patients with Total Joint Replacements versus a Control Group

**DOI:** 10.3390/medicina58060738

**Published:** 2022-05-30

**Authors:** Farhana Rahman, Felice Femiano, Patrick J Louis, Chung How Kau

**Affiliations:** 1Department of Orthodontics, University of Alabama at Birmingham, Birmingham, AL 35294-0012, USA; farhana.rahman@uab.edu; 2Dipartimento Multidiscilinare di Specialita Medico-Chirurgiche e Odontoiatriche, University of Campania “Luigi Vanvitelli”, 81100 Caserta, Italy; felice.femiano@unicampania.it; 3Department of Oral and Maxillofacial Surgery, University of Alabama at Birmingham, Birmingham, AL 35294-0012, USA; plouis@uab.edu

**Keywords:** jaw tracking, joint replacements, orthodontics, orthognathic surgery, jaw movements

## Abstract

*Background and Objectives*: One form of treatment for degenerative temporomandibular joint diseases such as osteoarthritis, rheumatic arthritis, TMJ ankylosis, and condylar resorption is total joint replacement. The aim of this study was to examine the function of the temporomandibular joint after prosthetic joint replacement. *Materials and methods*: Fifteen patients with unilateral or bilateral TMJ total joint replacements and 15 healthy controls were evaluated via a SICAT JMT+ device. This non-invasive system measures 3D position and linear movements in all degrees of freedom and allows undisturbed functional mandibular movements to provide a quantitative evaluation. In addition, a TMJ questionnaire consisting of the subjective symptoms was also obtained. To date, no similar studies have been cited in the literature. *Results*: Mandibular movements after prosthetic joint replacement were recorded during opening, closing, protrusion, and lateral excursive movements and were all significantly decreased compared to those of controls. In the treatment group, the maximum incisal opening was 33.46 ± 5.47 mm, left lateral movement was 1.91 ± 2.7 mm, right lateral movement was 1.74 ± 1.74 mm, and protrusive movement was 2.83 ± 2.05 mm. The *p*-value comparison study and control group indicated significant difference (*p* < 0.0001) between the two groups. The study group stated a high level of satisfaction with the total joint replacement. *Conclusion*: Within the limitations of the study, the following conclusions can be drawn: (1) TMJ replacement patients showed significantly limited jaw movements compared to the control group; (2) a small percentage of TMJ replacement patients still present low levels of pain but improved chewing ability and quality of life.

## 1. Introduction

The temporomandibular joint (TMJ) is a unique synovial joint, consisting of the articular disc, mandibular fossa of the temporal bone, condyle of the mandible, fibrous capsule, synovial fluid, and ligaments. It is located below the posterior end of the zygomatic arch and in front of the external acoustic meatus. Different mandibular movements such as opening, closing, protrusion, retrusion, and lateral excursions are carried out by the TMJ. It withstands various forces for mastication, deglutition, and phonation [1].

Temporomandibular joint disorders (TMDs) are multifactorial and characterized by pain, clicking, and difficulty with mouth opening and mastication. Studies have suggested that 41% of the population reported at least one symptom and 56% showed one clinical sign [1]. However, TMD is currently the most prevalent source of orofacial pain among 12% of the adult population in the United States. Females are 10 times more prone to it, with the majority between the ages of 18 and 45 years. Osteoarthritis, trauma, ankylosis, failure of the previous reconstruction, rheumatoid ar-thritis, and idiopathic condylar resorption are indicated for prosthetic joint replacement.

Several predisposing factors associated with temporomandibular joint degenerative diseases are gender (females), nutrition, genetics, oral habits (bruxism) and acute and chronic overload. Juvenile idiopathic condylar resorption has also been recently reported to cause significant destruction of the joint in the adolescent years [2,3,4,5,6]. The diagnosis of patients with TMJ disorders is performed by clinical and radiographic investigation [7]. Although panoramic and lateral cephalograms are routinely taken, it has been suggested that cone beam computed tomography (CBCT) is considered the imaging modality of choice to provide more information on spatial and temporal changes in the condylar head [8,9].

Management of the degenerative diseases of TMJ is complex and controversial. It can be conservatively managed with occlusal splints to reduce the load of the condyle and relieve TMJ pain and discomfort. Sansare et al., in a systematic review, reported no relapse in 24–27 months [10]. Orthognathic surgery alone had mixed results, with some studies showing improvements [11] but others a high rate of relapse (83.3–100%) over a follow-up of 1–39 months [10]. Papadaki et al. suggested performing orthognathic surgery only when the ICR is inactive. Condylectomy and total prosthetic joint reconstruction of the TMJ have been widely used to overcome the limitations of the biological and mechanical adaptability of the TMJ. Many case reports have been cited, but few well-controlled studies have been reported [12,13,14]. 

Custom-made total joint prostheses were developed in 1990 by Techmedica Inc. Camarillo, CA, USA, and are currently manufactured by TMJ Concepts, Inc., Ventura, CA, USA [15]. These CAD/CAM devices (computer-assisted design/computer-assisted manufacture) are designed to fit the specific anatomical requirements for each patient. In 2000, a stock TMJ replacement device by Zimmer Biomet, Jacksonville, FL, was introduced and found to be clinically useful [16]. Zimmer Biomet also manufactures a custom TMJ replacement device, only available outside the USA since it is not yet FDA-approved. Long-term studies support both stock and custom TMJ replacement [17]. Using alloplastic joints has the benefits of providing counterclockwise rotation and minimizing the risks of post-operative complications. Some authors showed excellent long-term stability and reduction in TMJ symptoms with total joint replacement over 5–12 years, but their studies lacked a large sample size. A systematic review by Sansare et al. revealed the need for long-term studies on the outcome of management of ICR with total joint replacement to establish a proper treatment guideline [10].

The normal adult mouth opening ranges from 38 to 50 mm, the normal lateral excursions from 8–10 mm, and the normal protrusions from 8 to 12 mm. However, in patients after total joint replacement, there is an increase in the maximum incisal opening compared with their pre-operative condition, but their jaw motion is significantly decreased compared to normal. Several case reports showed successful treatment of a case of bilateral idiopathic condylar resorption and anterior open bite with maxillary osteotomies and condylar replacement with TMJ concept custom condyle fossa (Figure 1) [12,13].

Three-dimensional jaw tracking studies have been conducted before. Wojczynska and co-workers used dynamic stereometry to assess the TMJ movements of patients in real time [18]. Baltali and co-workers also used jaw tracking in patients who had undergone hemi-joint reconstruction [19]. Furthermore, a working group led by Lindson and co-workers performed various jaw examinations on patients with both unilateral and bilateral TMJ issues [20,21]. SICAT JMT+ (Jaw motion tracking) (SiCAT, Cologne, Germany) is a sophisticated and reliable device (Figure 2) to assess temporomandibular joint movements, including opening–closing, lateral excursion, and protrusion. This ultrasound-based system converts multiple acoustic signals into spatial information, which records mandibular movements. Furthermore, other authors have showed that this system is capable of measuring 3D position and linear movements in all degrees of freedom and allows undisturbed functional mandibular movements [22]. The same system has also been used to evaluate the TMJ function of patients after orthognathic surgery [23].

Several studies have reported on the measurement of TMJ function after total joint replacement using a 3D jaw tracking device [18,19,20,24]. Therefore, the purpose of this study was to quantitatively assess patients who have undergone total TMJ replacement using a 3D jaw tracking device and evaluate the condition of the jaws. 

To date, no studies have used a jaw tracking device to evaluate post-operative joint function in patients who had total joint replacements versus a control group. The working null hypothesis was that there is no difference in range of motion and TMJ symptoms for patients undergoing total joint replacement versus a control group.

## 2. Materials and Methods

### 2.1. Study Design

A quantitative assessment of patients who had undergone TMJ replacement was conducted and compared with normal individuals to determine the differences in TMJ function and subjective symptoms. QoL and satisfaction were measured subjectively using a visual analogue scale.

### 2.2. Sample

Institutional Review Board approval (IRB-30000338) was granted for this research project; the committee consisted of the Director of UAB IRB and Board members who were not dentists. The database inquiry was initiated by screening the Department of Oral and Maxillofacial Surgery and the orthodontic department’s electronic patient record databases from 1993 to 2017. The inclusion criteria for the patients were as follows:(1)Anterior open bite (AOB);(2)Class I or II malocclusions;(3)Subject age range 18–66 years;(4)Patients who had been diagnosed with condylar resorption and underwent joint replacements;(5)Patients with a record in the Department of Oral and Maxillofacial Surgery or orthodontic department.

The exclusion criteria for the patients were as follows:(1)Presence of craniofacial syndromes;(2)Did not wish to participate;(3)Age under 18 years;(4)Presence of craniofacial deformities;(5)Association with any orofacial trauma.

### 2.3. Control Group

The control group consisted of subjects without any significant TMJ disorder. This study excluded patients that presented with any form of pain to the TMJ complex. These subjects (Table 1) were composed of residents and clinical staff of the orthodontic clinic of XXX with no known TMJ disorders according to the diagnostic criteria of significant TMD. Patients who filled in the subjective questionnaire and had pain in the joints were excluded from the study.

### 2.4. Informed Consent

Consent for the patient to participate took place in a closed treatment room in which the privacy of the patient and their treatment was maintained. Written and verbal informed consent was obtained. The consent form clearly stated the purpose of the study, a description of the procedure, the risk and benefit of the study, and the ability to discontinue the study at any point without penalty. 

Both the study group and the control group were provided with a previously cited self-report questionnaire based on TMJ pain (nature of pain, aggregating and relieving factors), clicking and popping, crepitus, locking, morning stiffness, and chewing ability [23,25,26] (Table 2). Pain perception was reported by the patients using the visual analogue scale (VAS). The VAS is graded from 0 (“no pain”) to 10 (“worst pain”). The muscles of mastication were examined; any muscular discomfort and joint crepitus or clicking were recorded.

TMJ incisal trajectories and movements (lateral excursion, protrusion, and opening–closing) were measured with SICAT JMT+ (Figure 3). This protocol was performed in both control and study subjects.

The SICAT JMT+ system has been previously reported on and consists of a face bow, a lower jaw sensor, a SICAT Fusion Bite Tray, a SICAT Fusion Bite adapter, a para-occlusal T-attachment, a SICAT JMT+ basic unit, and SICAT JMT+ application software [22,23,27] (Figure 2). The face bow has a nose pad to locate the face bow and a rear headband to secure it to head. Six ultrasonic microphones receive four ultrasound transmitters in the lower jaw sensor. The SICAT JMT+ basic unit relates to both the receiver and transmitter to evaluate the recording. The para-occlusal attachment of the ultrasonic transmitter attaches to the patient and blocks occlusal bite relationships. It was adjusted to the lower jaw dental arch, supplemented with autopolymerizing composite to the bending part of the T-attachment and adapted and hardened to the tooth surfaces. Excess and sharp material was removed. As a result of this procedure, the functional movement of the jaw in the occlusion was undisturbed since the maxillary teeth were not in occlusion. The measurement sensor technology consisted of a receiving sensor and a transmitting sensor.

### 2.5. Jaw Tracking Procedure

The upper jaw sensor was positioned stably on the patient’s head; the headband did not stretch the skin in the forehead area. The Fusion Bite Tray with the impression was positioned in the patient’s mouth and it was checked that the patient is biting into the right position. The impression material (Blue Mosse VPS, Patterson Dental, Edgewood, NY, USA) was a polyvinylsiloxane material (PVS). The T-attachment was placed in the patient’s mouth. This procedure was crucial to ensure that the data captured in the CBCT were accurately matched to the SICAT Device. The SICAT JMT+ software was then started and prepared for measurement (Figure 3). 

The lower jaw sensor technology was fitted with a special locking mechanism for fixing it to the attachment. After attaching the lower jaw sensor to the SICAT Fusion Bite Tray and clicking “record”, the software guided the program throughout the whole calibration sequence. The SICAT JMT+ lower jaw sensor was attached to the para-occlusal T-attachment and thereafter the “record” function was activated within the software. Subsequently, the SICAT Fusion Bite Tray was removed, and the sensor was mounted on the attachment so that the process of functional analysis could begin. Patient jaw movements, including jaw opening and closing movements, lateral excursive movements, and forward movements, were recorded. Recordings were repeated twice to validate the consistency of the jaw movements. In the case of large procedural discrepancies, the highest recording was accepted. The collected data were stored in an Excel (Microsoft, Redmond, WA, USA) spreadsheet, with the only identifiable patient information being the medical record number. The Excel spreadsheet was password-protected. Only the researchers participating directly in the study had access to this information.

### 2.6. Statistical Analysis

The Wilcoxon test was used to compare the continuous variables in two-sample data. The Wilcoxon test is a non-parametric statistical test used to compare two samples that are related to each other. It was used in this study as an alternative to the paired Student *t*-test because the distribution of the difference between the two samples could not be assumed to be normal. Moreover, this test is valid for small sample sizes. In addition, Fisher’s exact test was also used to evaluate the distribution of the variables in the total joint replacement patients and the control group. Student’s *t*-test was used to compare the means and standard deviations between the two groups and determine whether there is a statistically significant difference. 

## 3. Results

### 3.1. Sample Size

Fifteen prosthetic joint replacement patients were recruited for the study. The mean age of participants was 45.47 years for the study group and 39.61 years for the control group. All patients in the study group were females and the age range was 20–66 years. Twelve subjects had double TMJ replacement whereas two had a right joint replacement and one had left joint replacement. The study group had undergone either unilateral or bilateral TMJ replacement, performed 2–20 years before. The mean of the jaw replacement was 2.6 years, with the median at 2.5 years. The control group consisted of 15 healthy female individuals aged between 24 and 57 years. 

### 3.2. Clinical Evaluation

There were significant symptoms of TMD such as pain in the joints, incidence of locking, stiffness, clicking–popping, crepitation, constant pain, pain during movement, muscular pain, headaches, and migraines noted among two groups (Table 2). In addition, the VAS pain score was 1.9 ± 1.8 in the study group, which was significant when compared to the control group. The study subjects mentioned that their quality of life became better after the total joint replacement surgery. Furthermore, they mentioned that their chewing ability and mouth opening also improved in contrast to their pre-operative condition (Table 2).

### 3.3. Jaw Movement Evaluation

The mean values and standard deviations of maximal incisal opening (MIO), right and left lateral excursive movements, and protrusion are described in Table 3. The MIO was 33.5 ± 5.5 mm for the treatment group and 46.6 ± 8.2 mm for the control group. The *p*-value comparing the two groups was <0.0001 (Table 3). The left lateral movement was 1.9 ± 2.7 mm in the study group and 9.7 ± 1.4 mm for the control group. The *p*-value comparing the left lateral movement between the two groups was <0.0001. The right lateral movement was recorded at 1.7 ± 1.7 mm in the study group and 9.7 ± 1.6 mm in the control group. The *p*-value was <0.0001 among the two groups. The protrusive movement was found to be 2.83 ± 2.05 mm in the study group and 9.9 ± 1.5 mm in the control group. The *p*-value was <0.0001, indicating a significant difference between the control and study groups.

## 4. Discussion

This study represents one of the few studies that compares subjects after joint replacement using a 3D jaw tracker device. The success of the management of TMJ degenerative diseases depends on the functional and esthetic results and long-term stability [28,29]. The application of different approaches may result with variable relapse rates and short-term follow up data have led to poor understanding of the treatment outcomes. Wolford proposed reposition and stabilization of the disc with a mini anchor at the posterior part of the condyle. Over a period of 1.5–2 years, no relapse was shown [30]. The next successful treatment approach was condylectomy with a costochondral graft, with no relapse until 12 months [31]. However, some studies showed excessive growth of the costochondral graft, which resulted in the deviation of the chin, mandibular prognathism, and ankylosis [28]. Orthognathic surgery had the highest relapse rate with a short-term follow-up due to an increased rate of condylar resorption [32,33,34,35]. In 2015, Sansare et al. reported that there is a lack of evidence for treatment management and further studies with long-term follow-ups are needed to enable a proper treatment guideline [10]. A recent study reported that jaw trackers did not detect differences in TMJ function except some clinical symptoms after orthognathic surgery [23]. It was interesting to also note that the incisal movements were very similar in a reported study by Linsen [20].

In this study, the null hypothesis had to be rejected. All the subjects in our study group had undergone reconstruction of the TMJ with a follow-up of 2 to 20 years. Alloplastic total joint replacement was performed with TMJ Concept, which is a custom-made prosthesis consisting of fossa and mandibular components [36,37,38,39]. The fossa is constructed from a custom-made titanium sheet with a welded mesh that interfaces with the dense articulating surface, and the mandibular component is constructed with two basic materials: a Ti6Al4 V alloy coated with a Co–Cr–Mo alloy head. Many patients who had been treated at the Oral and Maxillofacial Surgery Clinic were referred from far away due to the complexity of the management of condylar resorption. Initially, 40 participants were contacted from the Oral and Maxillofacial Surgery database, but only five patients showed up for the study. It was anticipated that due to the lack of pain and functional problems, they were unwilling to come for the follow-up visits. The other 11 patients were chosen from their follow-up appointment and might have had some issues with their joint and needed a follow-up intervention. Twelve subjects had double TMJ replacement whereas two had right joint replacement and one had left joint replacement. All the patients in the study group mentioned that they sought treatment due to functional problems and occlusal instability. Fourteen subjects had received orthodontic treatment. 

The pain pathway in temporomandibular disorders is complicated and involves both biomedical and biopsychosocial factors. According to Okeson et al., patients can often suffer from neuropathic, neuropsychiatric, or myogenic pain or other underlying medical conditions that can influence the pain perception [1]. In our study, although some patients reported subjective perception of chronic pain, they added that it was more manageable with medication compared with the pre-operative pain. Thirteen among the fifteen subjects mentioned that the pain was below 2 on the VAS. Two subjects scored their pain as 5—dull chronic pain and manageable with pain medication. Interestingly, one subject in the control group scored their pain as 9 with an MIO of 30 mm, reduced lateral excursive movements and clicking, crepitus, and constant pain on the left side. Studies showed that only 5% of adults perceived a need for treatment among the 10–15% of adults who had the symptom of pain [40]. However, there was no statistically significant difference in VAS pain score between the study and control groups. 

In the literature, it is reported that the ideal maximum incisal opening in females is 48 ± 5 mm, which is highest at a young age and reduces with increasing age [41]. The mean age of our control group was 30 ± 10 years, whereas the mean age of our study group was 45 ± 16 years, indicating a large variation between the study group and the control group. According to Tzanidakis et al., 71% of patients reported improved pain and 61% patients had an improved range of incisal opening following joint replacement [42]. The expected MIO after TMJ replacement surgery is 30 to 35 mm [43]. In our study, the mean MIO was 33.4 ± 5.47 mm within the follow-up period of 2–20 yrs. The MIO was reported to be increased significantly post-operatively. Pre-operatively, the mean range of the MIO was 22–25 mm. Thus, the improvement in the MIO was significant. A recent study demonstrated no significant difference in MIO between control and orthognathic surgery groups [23]. Wolford et al. showed that the average post-operative MIO following TMJ replacement was 26 mm at 6 months, 26 mm at 12 months, 32 mm at 18 months, and 35 mm beyond 18-month follow-up [39]. Two subjects in this study required additional TMJ surgical procedures following reconstruction with the total joint prosthesis for the removal of fibrotic tissue. Pain and limitation of the TMJ are common problems noticed due to fibrosis and reactive bone around the prosthesis. Mercuri et al. reported a lower success rate for reducing pain among patients who had had more than nine previous surgeries [37]. However, Wolford et al. reported that the improvement was higher in multiple-surgery patients. Thirty percent of our sample had undergone multiple TMJ surgeries due to the lack of proper management guidelines before coming to the Department of Oral and Maxillofacial Surgery of UAB. Therefore, a possible explanation for not recording a higher MIO could be the small sample size and lack of willingness for patient participation in the study. 

In our study, lateral excursion movements were decreased in all patients and significantly different to the control group. In a previous study, it was reported that the average lateral excursion was 2.1 mm, and after total joint replacement, the average lateral excursion was decreased to 1.7 mm [39]. This finding is consistent with ours. This outcome could possibly be explained by the detachment of the lateral pterygoid muscle after the condylectomy procedure and subsequent fibrosis around the TMJ prostheses. In a previous study, Wolford et al. showed that even after reattachment of the lateral pterygoid muscle to the neck of the prosthesis by permanent suture, there was no improvement of the lateral excursive movements [44].

Our study group showed significantly decreased protrusive movement at 2.8 ± 2.1 mm compared to the control group, which was 9.9 ± 1.5 mm. A prospective cohort study reported a range of protrusive movement of 2–3 mm in patients with total joint replacement [18]. The possible reasons could be the lack of lateral pterygoid, hypomobility of the masticatory muscles, and the presence of scar tissues from previous surgical procedures. Using a mandibular simulator, it was observed that in cases where the MIO was not limited by the replacement, laterotrusion and protrusion were vastly impaired [45].

SICAT JMT is an electronic recording system based on 3D ultrasound measurement that converts the propagation times of multiple acoustic signals into spatial information. Therefore, this system is capable of recording the mandibular movements in all degrees of freedom in the case of control and study groups.

The majority of our subjects mentioned that their masticatory efficiency significantly improved. Despite some chronic pain and discomfort in a few patients, all agreed that their quality of life also improved remarkably. 

Our result suggested a significant difference in the TMJ movements, such as opening–closing, protrusion, and lateral excursive movements, compared to the control group. Within the limits of the subject groups, it seems that even though TMJ replacements may improve symptoms, limited jaw movements can still persist. One possible explanation could be that the majority of the sample (68.75%) was collected from scheduled patient follow-up appointments where the patient presented with some subjective pain or joint tenderness. Another possibility could also be the range of the post-operative follow-up, which was 2–20 years. 

One of the limitations of this study is the sample size. No sample size calculation was carried out due to the lack of previously reported data using 3D jaw tracking. Due to the novelty of the management of TMJ replacement with customized prosthesis, the study group was concise, and the patients recalled had to come from a long distance. As a result, many patients refused participation in the study, and many have even been shifted to another state. In addition, those that participated in the study may have either good or bad outcomes that could possibly skew the dataset. In addition, many of the readings were taken at the incisal level rather than the condylar level. This limitation was due to the sensors being attached only in the incisor region of the mandible. Lastly, due to the retrospective nature of the study and the length of time of treatment of such procedures, the patients were not assessed pre- or post-operatively by an independent examiner. In addition, the data describing masticatory function and QoL need further evaluation.

We hope to be able to further increase the sample size and to add other groups of interventional patients (e.g., orthognathic surgery) using the device protocol mentioned in this study.

## 5. Conclusions

It has been shown that there are differences in the outcomes of jaw movement after total joint replacement. However, the symptoms associated with TMD seem to be similar to the controls. Within the limitations of the study, the following conclusions can be drawn:TMJ replacement patients showed significantly limited jaw movements compared to the control group. These limitations were present in all range of motion movements.A small percentage of TMJ replacement patients still presented low levels of pain but an improved chewing ability and quality of life. This represented an overall improvement in the symptoms experienced by patients.

## Figures and Tables

**Figure 1 medicina-58-00738-f001:**
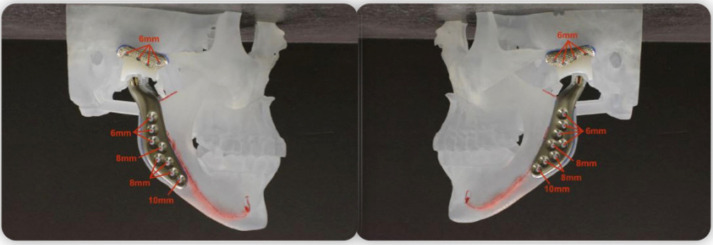
TMJ concept for total joint replacement of fossa and mandibular component.

**Figure 2 medicina-58-00738-f002:**
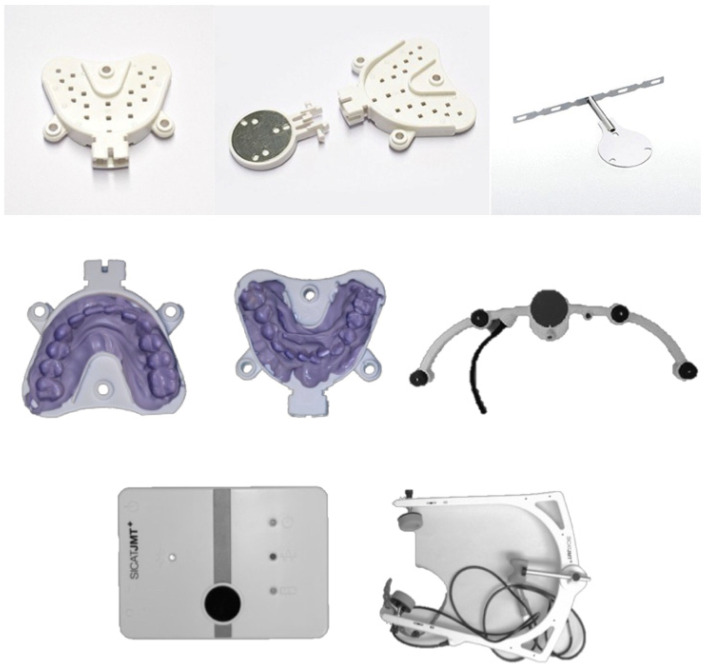
T-attachment, Fusion Bite Tray, lower jaw sensor, facebow with nasion support, neckband, and headband.

**Figure 3 medicina-58-00738-f003:**
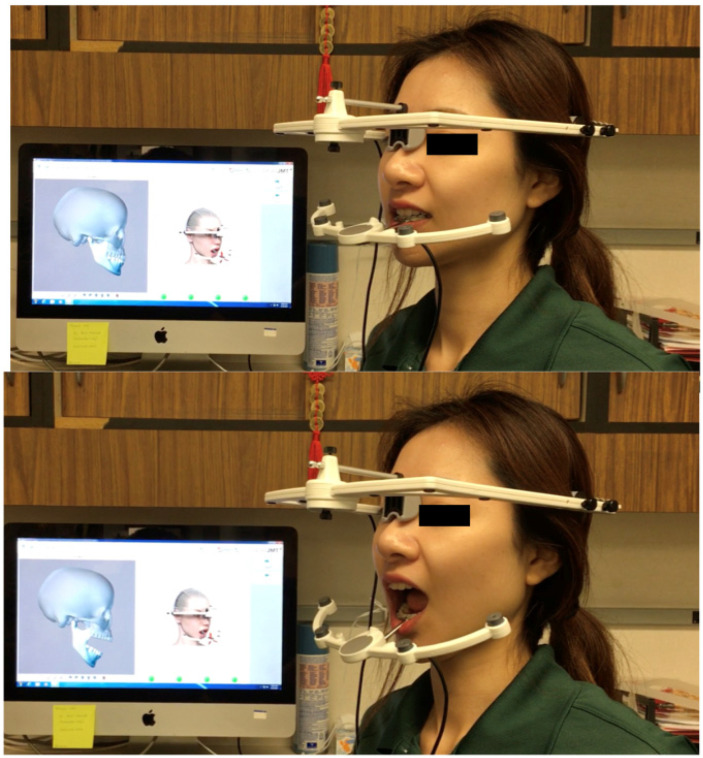
Extra-oral photos with SICAT JMT+. The maxillary mouthpiece was positioned to orientate the CBCT and maxillary position (not seen). Mandibular tracking device alone.

**Table 1 medicina-58-00738-t001:** Demographic data of sample number, age, and gender. Twelve patients had total joint replacement <5 years.

Variables	Number	Age (years)	Gender
Total joint replacement group	15	45.4 ± 16.7	15 females
Control group	15	39.6 ± 9.6	15 females

**Table 2 medicina-58-00738-t002:** Subjective symptoms on the sides of jaw examined and *p*-values of significance.

Outcome Comparisons	Total Joint Replacement (n = 15)	Control (n = 15)	*p*-Value
Clicking (Reported) Right	1 (6.7%)	0 (0.0%)	0.0996
Clicking (Reported) Left	1 (6.7%)	0 (0.0%)	0.0996
Clicking (On exam) Right	0 (0.0%)	0 (0.0%)	1.000
Clicking (On exam) Left	0 (0.0%)	0 (0.0%)	1.000
Popping Left	0 (0.0%)	1 (6.7%)	0.0996
Crepitation (Reported) Right	1 (6.7%)	1 (6.7%)	1.0000
Crepitation (Reported) Left	0 (0.0%)	0 (0.0%)	1.000
Crepitation (On exam) Right	1 (6.7%)	0 (0.0%)	1.000
Crepitation (On exam) Left	0 (0.0%)	0 (0.0%)	1.000
Constant pain Right	1 (6.7%)	0 (0.0%)	0.041
Constant pain Left	1 (7.1%)	0 (0.0%)	0.041
Pain in movements Right	3 (20.0%)	0 (0.0%)	0.011
Pain in movements Left	3 (20.0%)	0 (0.0%)	0.011
Stiffness	3 (20.0%)	0 (0.0%)	0.011
Muscular pain	1 (6.7%)	0 (0.0%)	0.041
Headaches	2 (13.3%)	0 (0.0%)	0.011
Migraines	1 (6.7%)	0 (0.0%)	0.012
Bruxism/clenching	3 (20.0%)	1 (6.7%)	0.778
Chewing ability better	15 (100%)	NA	
VAS pain score	1.9 ± 1.8	0.00	0.012 *

* Among the 15 subjects, 13 mentioned that the pain was below 2 in the VAS scale. Two subjects scored pain as 5; dull chronic pain and manageable with pain medication. One subject in the control group scored pain as 9.

**Table 3 medicina-58-00738-t003:** Range of mandibular movements measured in millimeters. Values in brackets are the median scores.

Mandibular Movements	Total Joint Replacement Group (Median Values)	Control Group (Median)	*p*-Value
Left lateral	1.9 ± 2.7 (1.5)	9.7 ± 1.4 (9.2)	<0.0001
Right lateral	1.7 ± 1.7 (1.2)	9.7 ± 1.6 (9.3)	<0.0001
Protrusion	2.8 ± 2.1 (2.3)	9.9 ± 1.5 (9.3)	<0.0001
Opening (MIO)	33.5 ± 5.5 (36.2)	46.6 ± 8.2 (46.1)	<0.0001

## Data Availability

Original data maybe requested from the corresponding author.

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
