# Peer review of "An Evaluation of Jaw Tracking Movements in Patients with Total Joint Replacements versus a Control Group"

_medicina, 2022, doi:10.3390/medicina58060738_

Round 1
Reviewer 1 Report
The study seems very interesting and genuine. However, the authors should address the following points:
* The abstract should be non-structured and word limit should be applied based on the journal's guidelines. Also, the current gap in literature should be reflected in the abstract using a short statement.
* The conclusion section of the abstract should be summarized without numbering.
* In the line 72, the authors ended up the sentence with "minimize the risks." the sentence should be completed (i.e. risk of what?).
* The authors should add the null hypotheses in the introduction section and reflect their findings whether to accept or reject these hypotheses in the discussion part.
* The research question and gap should be clarified in the introduction section.
* The authors should list all inclusion and exclusion criteria for clarity (not within the paragraph).
* The figures and tables should be impeded in the text in the areas they belonged to.
* How was the measurement technique calibrated? was there any reference point used for calibration? were the jaw movements operator-guided?
* The investigator should have averaged multiple doubted measurements instead of using the upper limits.
* Was there any statistical variation between the bilateral and unilateral joint replacement individuals?
* The discussion part should be fluent without subheadings.
* The authors should add future directions for more exploratory studies in this field.
* The conclusion part should be expanded and summarized in bullets.
Author Response
The study seems very interesting and genuine. However, the authors should address the following points:
Dear Reviewer - Thank you so very much for taking the time to review and for your valuable comments.
* The abstract should be non-structured and word limit should be applied based on the journal's guidelines. Also, the current gap in literature should be reflected in the abstract using a short statement.
We made the changes to the abstract as suggested.
* The conclusion section of the abstract should be summarized without numbering.
We made the changes as suggested.
* In the line 72, the authors ended up the sentence with "minimize the risks." the sentence should be completed (i.e. risk of what?).
Added the words post-operative complications
* The authors should add the null hypotheses in the introduction section and reflect their findings whether to accept or reject these hypotheses in the discussion part.
We added this information to both introduction and discussion.
* The research question and gap should be clarified in the introduction section.
We added the information
* The authors should list all inclusion and exclusion criteria for clarity (not within the paragraph).
Added this information as suggested
* The figures and tables should be impeded in the text in the areas they belonged to.
The journal will amend this as suggested.
* How was the measurement technique calibrated? was there any reference point used for calibration? were the jaw movements operator-guided?
No reference points were used and the results came directly from the electronic reading. Before the system can be used, a machine calibration exercise is conducted as per manufacturer's instructions.
* The investigator should have averaged multiple doubted measurements instead of using the upper limits.
We wanted to record the maximum possible range of motions and hence did not use the averages.
* Was there any statistical variation between the bilateral and unilateral joint replacement individuals?
We tried to do this but the numbers were too small for use.
* The discussion part should be fluent without subheadings.
We removed the sub-headings
* The authors should add future directions for more exploratory studies in this field.
We added this information.
* The conclusion part should be expanded and summarized in bullets.
We have done so.
Reviewer 2 Report
The authors aimed to quantitatively assess patients who had undergone total TMJ replacement using 3D jaw tracking device and evaluate the condition of the jaws. The study covers some issues that have been overlooked in other similar topics. The structure of the manuscript appears adequate and well divided in the sections. Moreover, the study is easy to follow, but some issues should be improved. Some of the comments that would improve the overall quality of the study are:
a. Authors must pay attention to the technical terms acronyms they used in the text.
b. English language needs to be revised.
c. Conclusion Section: This paragraph needs to be improved. Please also add some "take-home message".
Author Response
The authors aimed to quantitatively assess patients who had undergone total TMJ replacement using 3D jaw tracking device and evaluate the condition of the jaws. The study covers some issues that have been overlooked in other similar topics. The structure of the manuscript appears adequate and well divided in the sections. Moreover, the study is easy to follow, but some issues should be improved. Some of the comments that would improve the overall quality of the study are:
- Authors must pay attention to the technical terms acronyms they used in the text.
Thank you. We have made changes to the technical terms as suggested.
- English language needs to be revised.
We have made changes to many portions of the text.
- Conclusion Section: This paragraph needs to be improved. Please also add some "take-home message".
We made changes to the conclusion.